# What is the Visual Cognition Gap between Humans and Multimodal LLMs?

**Xu Cao**[1*], **Yifan Shen**[1*], **Bolin Lai**[2], **Wenqian Ye**[3], **Yunsheng Ma**[4], **Joerg Heintz**[1]
**Jintai Chen**[5], **Meihuan Huang**[6], **Jianguo Cao**[6], **Aidong Zhang**[3], **James M. Rehg**[1]
[1]Department of Computer Science, University of Illinois at Urbana-Champaign
[2]College of Computing, Georgia Institute of Technology
[3]Department of Computer Science, University of Virginia
[4]Digital Twin Lab, Purdue University
[5]HKUST (Guangzhou)
[6]Department of Rehabilitation Medicine, Shenzhen Children's Hospital
`{xucao2,yifan26,jrehg}@illinois.edu`

## Abstract

Recently, Multimodal Large Language Models (MLLMs) and Vision Language Models (VLMs) have shown great promise in language-guided perceptual tasks such as recognition, segmentation, and object detection. However, their effectiveness in addressing visual cognition problems that require high-level multi-image reasoning and visual working memory is not well-established. One such challenge is matrix reasoning – the cognitive ability to discern relationships among patterns in a set of images and extrapolate to predict subsequent patterns. This skill is crucial during the early neurodevelopmental stages of children. Inspired by the matrix reasoning tasks in Raven's Progressive Matrices (RPM) and Wechsler Intelligence Scale for Children (WISC), we propose a new dataset MaRs-VQA to evaluate the visual cognition capability of MLLMs and compare their performance with existing human visual cognition studies. Based on the training data of MaRs-VQA, we also finetune a baseline model Qwen2-VCog with multi-stage cognition reasoning annotations. Our comparative experiments with different baselines reveal a gap between MLLMs and human intelligence, highlighting the visual cognitive limitations of current MLLMs. We believe that the public release of MaRs-VQA and the Qwen2-VCog baseline model will drive progress toward the next generation of MLLMs with human-like visual cognition abilities. MaRs-VQA is available at huggingface.co/datasets/IrohXu/VCog-Bench. The training code of Qwen2-VCog is available at github.com/IrohXu/Cognition-MLLM.

## 1 Introduction

**Matrix reasoning** is a crucial ability in human cognition. It is used in non-verbal, culture-reduced intelligence assessments because it minimizes the influence of acquired knowledge and skills (Jensen, 1998; Jaeggi et al., 2010; Laurence & Macedo, 2023). Common matrix reasoning problems consist of images with simple symbols governed by underlying abstract rules (Małkiński & Mańdziuk, 2023) (see Figure 1). Participants have to identify and comprehend the rules based on a few provided patterns, and then reason about the next pattern following the same rules. Matrix reasoning is an important reflection of many fundamental capabilities of human visual cognition, such as processing speed and working memory, that emerge in the early stages of children's neurodevelopment (Gentner, 1977). It is also included in many assessment methods for fluid intelligence tests such as Wechsler Intelligence Scale for Children (WISC) (Wechsler & Kodama, 1949) and Raven's Progressive Matrices (RPM) (Raven, 2003).

---

*Equal contribution.

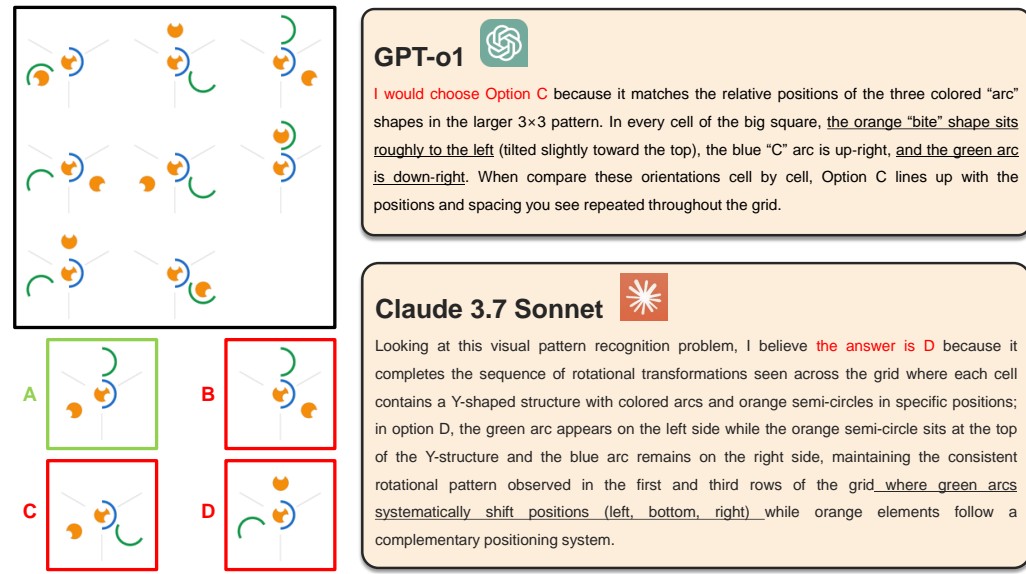

Figure 1: The example of the subpar performance of current state-of-the-art MLLMs (GPT-o1, Claude 3.7 Sonnet) on a simple matrix reasoning task used in MaRs-VQA (similar to cases in RPM and WISC). Both models can recognize the basic shapes in the provided patterns but fail to reason the next pattern.

In psychometrics, matrix reasoning tasks for children are specifically designed to assess visual reasoning abilities without prior specialized training. Children typically approach these tests relying solely on their general cognitive skills developed from everyday interactions with natural environments. This raises an intriguing question: Do Multimodal Large Language Models (MLLMs) exhibit visual cognitive capabilities similar to those of humans? MLLMs are trained on extensive general-domain data and have demonstrated the ability to generalize to unfamiliar tasks through in-context learning. However, current MLLMs still struggle with tasks that require advanced inductive reasoning, as evidenced by their poor performance on abstract reasoning tests such as the RAVEN IQ-test (Huang et al., 2024; Fu et al., 2024b; Yiu et al., 2024). The RAVEN IQ-test itself has notable limitations, including a relatively small dataset of only 50 samples (Huang et al., 2024), potentially introducing randomness and failing to robustly evaluate MLLMs' capabilities. Furthermore, it lacks comparative analyses with human performance, underscoring the need for more comprehensive and rigorous evaluation methods.

To address these gaps, we propose MaRs-VQA, a new visual question answering (VQA) dataset for general-purpose MLLMs, based on psychologist-certified matrix reasoning items with extensive sample diversity and rigorous human reference Chierchia et al. (2019). Unlike recent works, our benchmark uniquely offers: (i) direct comparison between generic MLLMs and humans on a scale an order of magnitude larger than previous VQA benchmarks and with richer stimuli types; (ii) rigorous psychological validity and baseline from human subject studies; (iii) explicit dual-modality annotation—for every question, both image and natural-language descriptions are provided for all options, enabling probing of language-vs-visual inference; (iv) full chain-of-thought (CoT) reasoning steps annotated, supporting deeper cognitive diagnosis and fine-tuning.

We also conduct thorough evaluation and comparison across 5 existing MLLMs (including their variants) and human performance under zero-shot inference setting (no prior knowledge) on MaRs-VQA and another abstract reasoning datasets RAVEN containing human studies. In our experiments, we observe that MLLMs with more parameters generally perform better on our benchmark, adhering to established scaling laws in a limited scope. However, even the largest open-source MLLMs and GPT-4o fall short of surpassing human performance in matrix reasoning tasks. In conclusion, our contributions are summarized as follows:

- We introduce a new matrix reasoning VQA dataset – MaRs-VQA, containing 1,440 image instances designed by psychologists, which is the largest dataset for matrix reasoning zero-shot evaluation.

- We conducted supervised fine-tuning (SFT) of Qwen2-VL using our annotated cognitive reasoning data. Our results indicate that fine-tuning enhances Qwen2-VL's accuracy on MaRs-VQA to match human performance levels; however, its generalization capabilities remain limited.

- Our thorough experiments qualitatively reveal the visual cognition gap between MLLMs and humans in matrix reasoning problems. We also show additional insights of deficiencies in MLLMs, which can inspire more future investigations in model design.

## 2 Related Works

**Cognitive Test of Large Language Models (LLMs)**    The rise of LLMs has aroused interest in exploring human-like AI in psychology and cognition (Ullman, 2023). Recent works tested LLMs' cognitive abilities in causal reasoning (Binz & Schulz, 2023), abstract reasoning (Xu et al., 2023b; Moskvichev et al., 2023; Jiang et al., 2024; Ahrabian et al., 2024), analogical reasoning (Webb et al., 2023), systematic reasoning (Hagendorff et al., 2023), and theory of mind (Strachan et al., 2024). Despite this success, only limited research has been conducted on the areas of MLLMs and visual cognition. Visual cognition involves the process by which the human visual system interprets and makes inferences about a visual scene using partial information. It is observed that while LLMs demonstrate a basic understanding of physical laws and causal relationships, they lack deeper insights into intuitive human preferences and reasoning Buschoff et al. (2023). Almost all existing visual cognition benchmarks focus on testing MLLMs' cognitive abilities in simple tasks (Zhou et al., 2023; Jassim et al., 2023), and ignore testing complex abstract reasoning and logical reasoning ability. Therefore, new and challenging benchmarks based on the theory of visual cognition are needed to assess and improve AI systems' capabilities for human-like visual understanding.

**Matrix Reasoning**    Matrix reasoning is often used to determine human intelligence related to visual cognition and working memory (Salthouse, 1993; Jaeggi et al., 2010; Fleuret et al., 2011) that is widely used by RPM (Raven, 2003; Soulières et al., 2009), WISC (Wechsler & Kodama, 1949; Kaufman et al., 2015) to evaluate human ability to detect the underlying conceptual relationship among visual objects and use reasoning to find visual cues. Early research indicated that deep learning models can be trained to solve simple matrix reasoning (Małkiński & Mańdziuk, 2022; 2023; Xu et al., 2023a; Małkiński & Mańdziuk, 2024) and compositional visual relation tasks (Fleuret et al., 2011; Zerroug et al., 2022; Liu et al., 2021). Several datasets and benchmarks are also proposed, such as RAVEN (Zhang et al., 2019), RAVEN-I (Hu et al., 2021), RAVEN-FAIR (Benny et al., 2021), CVR (Zerroug et al., 2022). However, these works have a key limitation. They overlook the fact that humans can solve such puzzles in a zero-shot manner, without explicit training on large-scale data. Recently, there are also some useful zero-shot visual reasoning inference datasets such as RAVEN-IQ (Huang et al., 2024), Visual Reasoning Benchmark (Zhang et al., 2024b), but all of them are limited by lacking rigorous human experiments as reference and conducting experiments on relatively small datasets without psychometrical validation.

**Vision-Language Models**    Researchers have been actively investigating the utility of Vision-Language Models (VLMs) for addressing vision reasoning tasks (Zellers et al., 2019; Bordes et al., 2024). These latest VLMs are constructed using a combination of the CLIP vision encoder, pretrained LLMs, and a connected adapter to align visual features with language space (Zhang et al., 2024a; Shao et al., 2024; Gupta & Kembhavi, 2023; Fu et al., 2024b). Notably, methodologies such as MiniGPT-4 (Zhu et al., 2023), InstructBLIP (Dai et al., 2024), LLaVA (Liu et al., 2024), CogVLM (Wang et al., 2023) underscore the significance of employing high-quality visual instruction tuning data. Nevertheless, current VLMs encounter challenges in adapting to high-resolution and visually complex images. These problems stem from the absence of a robust visual search mechanism (Wu & Xie, 2023),

| Dataset | Source | Sample | Instance | RGB image | Human Study | Psychological Validity | Open-source | VQA Annotation |
|---------|--------|--------|----------|-----------|-------------|------------------------|-------------|----------------|
| kosmos-iq50 (NeurIPS-23) (Huang et al., 2024) | RAVEN-IQ Test | | 50 | ✗ | ✗ | ✓ | ✗ | ✗ |
| Visual Reasoning Benchmark (COLM-24) (Zhang et al., 2024b) | Mensa Test, RAVEN, IntelligenceTest | | 241 | ✗ | ✗ | ✗ | ✗ | ✗ |
| MaRs-VQA (ours) | MaRs-IB Questionnaire | | 1,440 | ✓ | ✓ | ✓ | ✓ | ✓ |

Table 1: Comparison of recently released zero-shot matrix reasoning datasets to evaluate MLLMs.

few-shot reasoning (Guo et al., 2023), compositional understanding (Yuksekgonul et al., 2022) and the constrained visual grounding capabilities inherent in CLIP (Tong et al., 2024).

# 3 MaRs-VQA Dataset

The MaRs-VQA dataset is designed to evaluate the zero-shot abstract reasoning capabilities of MLLMs through various matrix reasoning VQA tasks. The images in MaRs-VQA are sourced from the questionnaires in Matrix Reasoning Item Bank, which is created by psychologists including 18 sets of abstract reasoning questionnaires (80 instances in each set) for non-verbal abstract reasoning assessment of adolescents and adults (Chierchia et al., 2019). Each item presents an incomplete $3 \times 3$ matrix of abstract shapes, requiring participants to identify relationships among the shapes. Then, we create VQA annotations in the images from all questionnaires. The comparison among MaRs-VQA and previous matrix reasoning benchmark datasets is shown in Table 1.

To transform the matrix reasoning problem into a VQA task, we define two option sets – image-based set and language-based set. In the image-based set, we provide four candidates to the missing patch in the question. We further diversify the modalities of our dataset to support the evaluation of different kinds of models. Specifically, the author teams annotate language-based descriptions for each option, forming language-based set. Each option annotation is formatted by GPT-4o to ensure consistency. In this process, we first manually design 10 formatted language-based sample pairs. These examples are then used as few-shot samples to query GPT-4o through in-context learning. The context generation system prompt guides GPT-4o to re-caption all annotations. After generating all samples, human annotators in the author team review each option again and revise the incorrect description. In Table 1, compared with other matrix reasoning datasets for MLLM's visual cognition evaluation, MaRs-VQA is the largest one with unique features on psychological validity, human study reference, VQA annotations.

# 4 Problem Statement

In this section, we introduce the evaluation pipeline of MaRs-VQA under multi-image reasoning setting.

Assume that the test set contains $n$ VQA samples, denoted as $\{(\mathbf{q}_i, \mathbf{x}_i, \mathbf{y}_i)\}_{i=1}^{n}$. $\mathbf{q}_i$ represents the question image showing the $3 \times 3$ matrix reasoning task. $\mathbf{x}_i = [x_i^1, ..., x_i^k]$ represents the images in the option set, where $k$ is the number of options. $\mathbf{y}_i$ is the answer of the matrix reasoning question. The inference pipeline can be formulated as:

$$\hat{\mathbf{y}}_i = F_\theta(\mathbf{q}_i, \mathbf{x}_i, \mathbf{x}_{sys}). \tag{1}$$

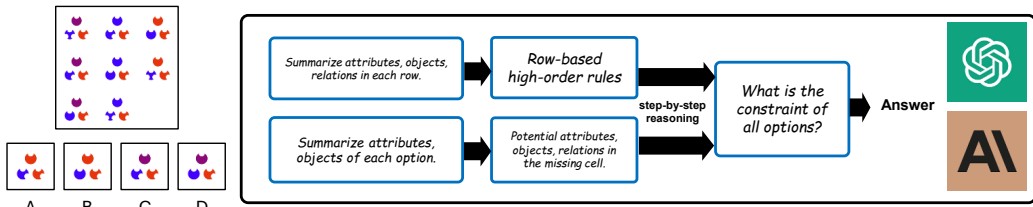

Figure 2: An overview of using CoT to solve matrix reasoning problem in MaRs-VQA. The left part is the model input, including a question image, multiple option images. The right part shows the step-by-step CoT for multi-image reasoning for GPT series and Claude series inference.

$\mathbf{x}_{sys}$ is the system prompt, including independent information about the matrix reasoning problem setting, step-by-step reasoning examples (optional), few-shot examples (optional), requirements for the output format. $\hat{\mathbf{y}}_i$ is the prediction result. $F_\theta$ is an autoregressive decoder in the MLLM for answer generation. It is defined as:

$$P(\hat{\mathbf{y}}_i|\mathbf{q}_i, \mathbf{x}_i, \mathbf{x}_{sys}) = \prod_{j=1}^{L} P(\hat{\mathbf{y}}_{i,j}|f(\mathbf{q}_i, \mathbf{x}_i), \mathbf{x}_{sys}, \hat{\mathbf{y}}_{i,<j}; \theta), \qquad (2)$$

where $f$ is the visual encoder and adapter layer, $L$ is the sequence length of answers and $\hat{\mathbf{y}}_{i,<j}$ is all answer tokens before $\hat{\mathbf{y}}_{i,j}$.

## 5 Methods

As we claim in previous sections, our initial goal is to complete the $3 \times 3$ matrix by finding the missing cell from multiple options by **zero-shot prompt engineering** under the same setting in human's matrix reasoning test. To this end, MLLMs have to deduce relationships across the other cells of the matrix and infer the missing cell accordingly. We use CoT prompt engineering to guide closed-source MLLMs solving this problem. To further promote related visual cognition foundation model, we also propose to use step-by-step structured reasoning annotations in MaRs-VQA to supervised finetune (SFT) Qwen2-VL with LoRA. Then we test the performance of Qwen2-VCog in both MaRs-VQA test set (in-domain) and a subset of RAVEN as Out-of-Domain (OOD) data. The observation in the experiment could reveal why this problem is hard for MLLM and highlight the gap between human intelligence and MLLM reasoning.

### 5.1 Multi-Image Reasoning via Chain-of-Thought (CoT)

Building on recent insights into systematic, language-based reasoning (Wei et al., 2022; Kojima et al., 2022), we propose a straightforward yet effective division of the reasoning workflow into two distinct and clearly tagged stages (<think> and <answer>), as illustrated in Figure 3. This design draws inspiration from methods such as OpenAI o1 (Zhong et al., 2024), LLaVA-CoT (Xu et al., 2024), and R1-V Chen et al. (2025) where each stage contributes a different level of abstraction to the overall inferential process:

- **Reasoning (<think>)**: It includes: (i) a concise overview of the task (e.g., 'examining a $3\times3$ grid puzzle and determining the missing cell'); (ii) a structured description of relevant visual elements (color, shape, position, etc.) that guide the reasoning; and (iii) a methodical analysis of the discovered pattern(s). Crucially, this step covers both rule identification (the model pinpoints how objects or elements follow consistent patterns) and option verification (each candidate option is tested against the identified rule).

- **Conclusion (<answer>)**: A single, succinct statement that specifies the best or correct choice among the provided options. No extra explanation is given here. It simply states which option is correct as the final answer.

This two-section format promotes a structured and transparent reasoning style by first concisely outlining the puzzle, then detailing all pertinent visual characteristics, followed by explicit discovery and testing of candidate rules, and finally isolating the single correct response. To further clarify the CoT processes in LLM-based reasoning, our method annotates each stage of the two-section format with dedicated tags, such as <think>...</think> and <answer>...</answer>. By explicitly marking the beginning and end of each reasoning stage, the model is guided to retain clarity and precision throughout the entire solution path. Unlike the traditional free-form CoT that allows the model to produce unconstrained self-talk, our approach enforces a well-structured methodology. A detailed template demonstrating this two-section CoT format is provided in our code repository.

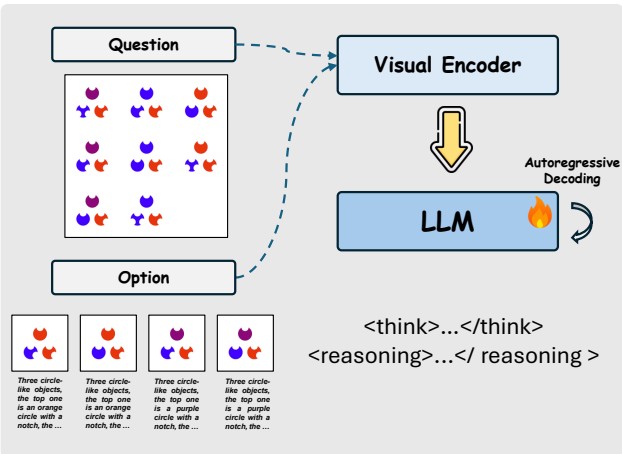

Figure 3: Supervised fine-tuning VLM to generate two-section format for matrix reasoning problem.

## 5.2 SFT for Vision-Language Model (VLM)

To enhance the reasoning capabilities of vision-language models (VLMs), we leverage the reasoning responses generated using the two-section CoT format as training data (in Figure 3). LoRA Hu et al. (2022) is employed to fine-tune Qwen2-VL, enabling efficient adaptation of the model to these structured reasoning tasks. Specifically, the step-by-step reasoning annotations in MaRs-VQA are used as supervision signals during fine-tuning. After training, the performance of the resulting model, Qwen2-VCog, is evaluated on MaRs-VQA test set for in-domain performance and a subset of RAVEN for Out-of-Domain (OOD) performance.

To further probe the contribution of explicit cognitive supervision, we conduct an ablation study by fine-tuning a variant of Qwen2-VL without the step-by-step reasoning annotations, in parallel to Qwen2-VCog (which was trained with full reasoning chains). This allows us to isolate the effect of reasoning supervision on both in-domain (MaRs-VQA) and out-of-domain (RAVEN) performance.

# 6 Experiments

## 6.1 Experimental Settings

**Datasets** For MaRs-VQA, We split 480 VQA samples as the test set and the rest of them is the training set for SFT. In addition to MaRs-VQA, we also select 560 VQA pairs serving as OOD samples from matrix reasoning dataset RAVEN (Zhang et al., 2019). Details of difference between MaRs-VQA and RAVEN are shown in Table 9 in appendix.

**MLLM Baselines** We selected the Claude 3 Sonnet, Claude 3 Opus, Claude 3.5 Sonnet (Anthropic, 2024), GPT-4V (OpenAI, 2023), GPT-4o (OpenAI, 2024b), GPT-o1 (OpenAI, 2024c), LLaVA-NExT (Liu et al., 2024), Qwen2-VL (Wang et al., 2024), InternVL-2.5 Chen et al. (2024) as the primary multi-image CoT baselines as they support multiple images input and can

| Method | Model Scale | Accuracy (%) ↑ | |
| --- | --- | --- | --- |
| | | MaRs-VQA (4-options) | RAVEN (8-options) |
| Random Select | - | 25.00 | 12.50 |
| LLaVA-NExT (Liu et al., 2024) | 7B | 16.88 | 14.29 |
| InternVL-2.5 Chen et al. (2024) | 8B | 20.00 | 13.21 |
| Qwen2-VL Wang et al. (2024) | 7B | 23.75 | 29.27 |
| Claude 3 Sonnet (Anthropic, 2024) | - | 23.22 | 13.39 |
| Claude 3 Opus (Anthropic, 2024) | - | 24.13 | 11.95 |
| Claude 3.5 Sonnet (Anthropic, 2024) | - | 24.28 | 15.36 |
| GPT-4V (OpenAI, 2023) | - | 33.13 | 15.63 |
| GPT-4o (OpenAI, 2024b) | - | 33.96 | 25.89 |
| GPT-o1 (OpenAI, 2024a) | - | 52.29 | 25.36 |
| Qwen2-VCog (Our SFT Baseline) | 7B | 72.71 | 31.96* |
| Human | - | **81.00** | **84.41** |

Table 2: Benchmarking experiments on multi-image reasoning in MaRs-VQA (in-domain) and RAVEN (OOD). zero-shot means only provide the model system prompt about the matrix reasoning task definition. CoT denotes the implementation in section 5.1. The results are averaged over three runs with three different random seeds. * of Qwen2-VCog denotes CoT performance without finetuning.

generate reasoning process. As Qwen2-VL (Wang et al., 2024) is the best open-sourced VLM for zero-shot inference in Table 2, we choose it as the main backbone to finetune our baseline Qwen2-VCog with MaRs-VQA training data.

**Human Baseline**   The human study results in Table 2 are reported from previous experiment results. The human subjects of RAVEN (Zhang et al., 2019) consists of college students from a subject pool maintained by the Department of Psychology. Only "easily perceptible" examples were used in the investigation. The human study results of MaRs-IB (Chierchia et al., 2019) are more rigorous. They are from 4 age groups ($N = 659$, aged 11–33 years). The accuracy for younger adolescents, mid-adolescents, older adolescents, and adults solving matrix reasoning in MaRs-IB are 61%, 68%, 73%, 81%. We use the adult result 81% in Table 2.

**Implementation**   For closed-source baseline models, we establish basic prompts to introduce the matrix reasoning problem setting, which serve as the system prompt for zero-shot inference. For open-source baseline models, we use the same system prompt settings across all models. Testing is conducted using two NVIDIA H100 GPUs for all models. All experiments are run with three different random seeds, and the results are averaged. We evaluate the results based on the accuracy of single-option matrix reasoning problems (Acc = Correct/Total), consistent with other VQA benchmarks (Lu et al., 2022; Liu et al., 2023).

## 6.2   Experimental Results

For all models in Table 2, we used multiple images as the input, including a question image and several option images, and guided the MLLMs to decompose the problem into predefined structures before generating answers based on all available information. We tested the latest closed-source models like Claude 3, Claude 3.5, GPT-4V, GPT-4o, and GPT-o1 for this task, as these models can generate step-by-step multi-image reasoning. In addition, we also compare them with small size open-sourced models like Qwen2-VL, InternVL-2.5. Our results show that even the state-of-the-art closed-source MLLMs GPT-o1 perform worse than humans in all matrix reasoning tasks.

After analyzing the reasoning outputs of current MLLMs, we identified three primary issues: (1) Limited Use of Visual Information: MLLMs struggle to directly utilize visual features during reasoning, rendering them insensitive to non-verbal spatial details, particularly evident when interpreting positional relationships within images. For instance, distinguishing among the options in Figure 1 is challenging for MLLMs using language alone. (2) Restricted Visual Working Memory: MLLMs exhibit limited visual working memory, leading to rapid loss of crucial visual information during text-based reasoning processes. (3) Integration

| Strategy | Accuracy (%) ↑ |
|---|---|
| GPT-4o+ CoT | 33.96 |
| GPT-4o+ CoT + 1-shot | 35.22 |
| GPT-4o+ CoT + 3-shot | 36.10 |
| GPT-4o+ CoT + 5-shot | 36.03 |
| GPT-4o+ multi-round CoT | 41.96 |
| GPT-4o+ multi-round CoT + 1-shot | 42.08 |
| GPT-o1 | 52.29 |

Table 3: Ablation on few-shot sample CoT and multi-round CoT for GPT-4o in MaRs-VQA. GPT-o1 still outperforms GPT-4o with different CoT strategy.

Challenges: Despite excelling at specific visual tasks like recognition, segmentation, and object detection, MLLMs encounter significant difficulties integrating these skills for high-level visual reasoning tasks. Further examples illustrating GPT-4o's failure cases and discussion are provided in the Appendix.

To address these shortcomings, we leveraged the reasoning annotations described in Section 5.1 to fine-tune Qwen2-VL using Low-Rank Adaptation (LoRA) as our baseline model for MaRs-VQA. After fine-tuning on the MaRs-VQA training set, the model achieved over 70% accuracy on the MaRs-VQA test set, nearing human adult performance. Additionally, its performance on the OOD RAVEN dataset improved from 29.27% to 31.96%. This observation shows that VLM could exploit shortcut features from matrix reasoning problem. It tends to perform well on the in-distribution test sets while slightly improving the out-of-distribution test performance. This indicates that there is still a major gap for current MLLMs to learn intended features in the abstract reasoning tasks. The simple strategy of SFT cannot entirely solve this task. This also demonstrates the value of our proposed benchmark in examining the out-of-domain nature of abstract reasoning tasks.

### 6.3 Ablation Study

In this subsection, we conduct ablation experiments to analyze how to improve the zero-shot performance of MLLMs on the matrix reasoning problem.

**Effect of CoT Strategy for Closed-source MLLMs**    Table 3 compares the different CoT strategy: raw step-by-step CoT with hint, CoT reasoning with few-shot sample and multi-round CoT reasoning. Few-shot samples are a small number of question-answer examples alongside the CoT system prompt. Multi-round reasoning employs the advanced multi-round CoT strategy (step by step option elimination). After each round, the model will reflect on the correctness of the answer and run the reasoning steps again if the answer is wrong. The results show that incorporating 1-shot and 3-shot samples gradually increases the accuracy of GPT-4o on MaRs-VQA from 34% to 36%. However, extending the number of examples to 5 does not yield further improvement. These findings suggest that while few-shot in-context learning helps the model better understand the matrix reasoning problem, it does not significantly enhance the MLLM's visual reasoning capabilities for these tasks. Additionally, using a multi-round elimination strategy improves accuracy from approximately 34% to 42%, but it is considerably slower than single-round CoT, and still cannot surpass GPT-o1 (52.29%) and human adults (81.00%).

**Effect of Reasoning Strings in SFT**    To more precisely quantify the impact of explicit step-by-step reasoning supervision in Supervised Fine-Tuning (SFT), we conducted an ablation study in which the reasoning strings were omitted from the MaRs-VQA training data. As reported in Table 4, the absence of these structured cognitive annotations led to a dramatic decrease in accuracy for Qwen2-VCog on both MaRs-VQA and RAVEN. Specifically, performance on MaRs-VQA dropped by nearly 18 percentage points, while the accuracy on RAVEN also decreased, albeit to a lesser extent. This result highlights that detailed reasoning guidance is not merely auxiliary, but instead serves as a crucial signal for the model to acquire transferable visual cognitive skills. Without such supervision, the model tends to rely on shallow pattern matching or spurious correlations, failing to generalize robustly even within the same task domain. These findings underscore the necessity of incorporating high-quality, stepwise annotations for training MLLMs to approach human-like reasoning on complex visual cognition benchmarks.

| Model | MaRs-VQA (%) ↑ | RAVEN (%) ↑ |
|---|---|---|
| Qwen2-VCog (with reasoning) | 72.71 | 31.96 |
| Qwen2-VCog (w/o reasoning) | 54.82 | 29.30 |

Table 4: Effect of reasoning strings in SFT.

**Vision vs. Language Bottleneck**   We further investigated whether the primary limitation of current MLLMs in matrix reasoning tasks arises from their visual perception modules or from downstream language-based reasoning. To disentangle these factors, we designed three input conditions: (1) providing only the question image and selecting options via CLIP similarity, (2) providing both the question and all option images (the standard VQA setting), and (3) supplementing the images with perfect human-annotated textual descriptions for both the question and options. As shown in Table 5, both GPT-4o and Qwen2-VCog perform at near chance level when deprived of option images, indicating a failure to extract the underlying visual rules from the question image alone. When option images are provided, accuracy increases substantially, especially for Qwen2-VCog, which demonstrates strong visual pattern matching. Notably, the addition of ideal textual descriptions yields only a marginal further improvement, suggesting that the language reasoning component is not the main bottleneck once high-quality visual features are available. These results collectively point to visual pattern extraction—rather than linguistic inference—as the principal limiting factor for MLLMs on abstract visual reasoning tasks, emphasizing the need for stronger visual encoders and better integration of visual working memory.

| Input Setting | GPT-4o (%) ↑ | Qwen2-VCog (%) ↑ |
|---|---|---|
| Question image only | 24.58 | 26.32 |
| Question image + Option images | 33.96 | 72.71 |
| Question image + Option images + Option Description | 36.46 | 71.43 |

Table 5: Vision vs. Language Bottleneck Analysis.

**Generalizability**   To ensure that the improvements observed from SFT on MaRs-VQA do not come at the expense of general visual-language understanding, we systematically evaluated both Qwen2-VCog and the original Qwen2-VL on a suite of standard multimodal benchmarks, including MME (Fu et al., 2024a), HallusionBench (Guan et al., 2024), POPE (Li et al., 2023), VQAv2 (validation set) (Goyal et al., 2017), SQA_Image (Lu et al., 2022), and SeedBench (Li et al., 2024). As summarized in Table 6, the two models exhibit nearly identical performance across all tasks, with only negligible fluctuations that are well within the range of experimental noise. This demonstrates that reasoning-focused finetuning on MaRs-VQA does not degrade the model's ability to perform generic vision-language tasks, nor does it induce catastrophic forgetting. Consequently, it is feasible to endow MLLMs with enhanced visual reasoning skills via SFT on MaRs-VQA, without sacrificing their broader applicability or real-world utility.

| Model | MME ↑ | Hallusion_Bench ↑ | POPE ↑ | VQAv2_Val ↑ | SQA_Image ↑ | SeedBench ↑ |
|---|---|---|---|---|---|---|
| Qwen2-VL-7B | 1666 | 53.83 | 88.84 | 79.88 | 83.73 | 69.16 |
| Qwen2-VCog-7B | 1680 | 55.73 | 88.71 | 79.45 | 83.29 | 68.87 |

Table 6: General VLM Benchmarks: SFT does not degrade generic ability.

## 6.4 Visualization

We also analyze the relationship between matrix reasoning accuracy and model scale in Figure 4. The figure illustrates the significant gap between MLLM's matrix reasoning performance and that of humans. This gap is substantial and suggests that simply increasing model size according to scaling laws will not be sufficient to bridge it.

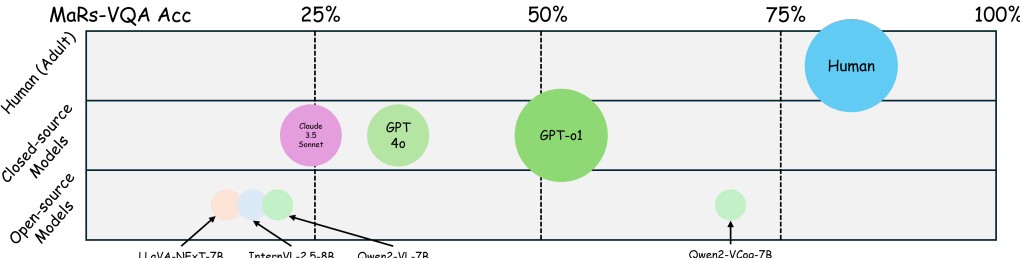

Figure 4: There is still a substantial gap between MLLM's (zero-shot CoT or SFT training) matrix reasoning capability and human's (zero-shot). Bubble size corresponds to the model size. As we don't know the exact size of closed-source MLLMs, we set all of them to the largest value by default. The model size of human refers to the number of neurons (86B) in human's brain (Voytek, 2013).

## 7 Discussion

In this work, we emphasize that zero-shot matrix reasoning is a crucial testbed for human-level intelligence, even though the developmental origins of this ability in children remain unclear. Remarkably, children as young as four can solve matrix reasoning problems without explicit training, highlighting the unique strengths of human visual cognition. Our long-term goals are twofold: (1) to rigorously evaluate how close MLLMs are to human-like cognitive abilities, as posed by Chollet (2019); and (2) to develop MLLM-powered agents capable of human-level zero-shot matrix reasoning, which could in turn generate novel assessment tools to help psychologists and pediatricians understand neurodevelopment.

Our findings reveal a persistent visual cognition gap between current MLLMs and humans, even as model scale increases. Detailed ablation and analysis suggest that the primary bottleneck lies in visual pattern extraction and working memory, rather than language reasoning. This gap has concrete implications for real-world applications: it limits the reliability of AI in dynamic or safety-critical environments (e.g., robotics, scientific discovery, education), where robust abstract visual reasoning and generalization are essential. Bridging this gap will require advances in both data and architecture, particularly in strengthening the visual encoders and multimodal integration of MLLMs. Our benchmark and insights lay the groundwork for future research toward more human-like, generalizable visual cognition in AI systems.

## 8 Conclusion

We introduce MaRs-VQA, a publicly available matrix reasoning Visual Question Answering (VQA) dataset specifically designed to evaluate the visual cognitive capabilities of Multimodal Large Language Models (MLLMs) and compare them with humans. Our findings indicate that state-of-the-art MLLMs, such as GPT-4o, Qwen2-VL, and InternVL-2.5, demonstrate foundational competence in matrix reasoning but continue to struggle with more complex or abstract scenarios, performing substantially below human levels. Supervised fine-tuning (SFT) with cognitively designed step-by-step reasoning annotations from MaRs-VQA can significantly boost in-domain accuracy, yet these models still fall short of human performance and generalize poorly to out-of-domains (OOD). Notably, humans achieve strong performance without any task-specific training, underscoring the inherent gap in visual cognition between humans and MLLMs. Our ablations further show that explicit reasoning supervision is crucial, and that vision—not language—remains the dominant bottleneck. Bridging this gap will require continued research and innovation in both model architecture and multimodal learning paradigms, ultimately advancing the visual cognitive abilities of future MLLMs.

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
