# OpenReview forum: "What is the Visual Cognition Gap between Humans and Multimodal LLMs?"
_colmweb.org/COLM/2025/Conference — COLM 2025_

### Official Review · Reviewer_1KjF · 2025-05-01

**Rating:** 5
**Confidence:** 4
**Ethics Flag:** 1

**Summary:**

This paper presents MaRs-VQA, a new benchmark for evaluating the visual reasoning capabilities of Multimodal Large Language Models (MLLMs) using data derived from the Matrix Reasoning Item Bank. The benchmark is designed around visual question answering (VQA) tasks inspired by classic matrix reasoning problems, with a particular focus on challenging MLLMs to capture abstract visual patterns. The authors propose a Multi-Round Chain-of-Thought (CoT) strategy to enhance reasoning across multiple visual inputs.

**Questions To Authors:**

Please refer to the weakness section.

**Reasons To Accept:**

- The proposed Multi-Round CoT strategy is an interesting addition that may inspire future adaptations of step-by-step reasoning in MLLMs.

**Reasons To Reject:**

- The dataset contribution is limited in originality: all the visual problems are repurposed from existing resources, leading to a marginally novel benchmark.
- The analysis lacks depth; the paper fails to provide novel insights beyond what prior work has indicated regarding the performance gap between MLLMs and humans on visual reasoning tasks.
- There is insufficient exploration of why the fine-tuned model shows major gains on the in-domain MaRs-VQA set but limited improvement on RAVEN—the discussion could have shed light on potential overfitting, domain shift, or symbolic reasoning limitations.
- The paper does not meaningfully interpret what the gap between MLLMs and human performance implies for real-world tasks or downstream applications—this would have been valuable for the COLM audience.

---

> ### Author Response · Authors · 2025-06-02
> **Response to Reviewer 1KjF (Part 1)**
>
> We thank Reviewer 1KjF for your encouraging comments, especially the suggestions on more explanation for the matrix reasoning domain shift problem. Firstly, we highlight our novelty and explain our main contribution and research workflow in detail. Secondly, we discuss our observation is aligned with common knowledge in shortcut learning and provide an additional experiment on controlling the option number of RAVEN to mitigate domain shift, which we hope allow raising post-rebuttal scores.
>
> **Question: The dataset contribution is limited in originality: all the visual problems are repurposed from existing resources, leading to a marginally novel benchmark.**
>
> Thank you for your feedback. We would like to clarify the novelty and contribution of our MaRs-VQA benckmark. While the foundational visual stimuli are adapted from the established Matrix Reasoning Item Bank (Chierchia et al., 2019), our primary contribution lies in the significant and novel transformation of these items into the MaRs-VQA benckmark. This process involved more than just repurposing images; we introduced several innovative elements crucial for evaluating and advancing Multimodal LLMs:
> 1\. **Novel VQA Framework and Annotation**: We are the first to convert these psychometric problems into a VQA format systematically. This involved creating language-based descriptions for all answer options (as detailed in lines 129-130 of our manuscript), providing an alternative input modality to the purely image-based options. This allows for a more diverse evaluation of MLLMs, testing their comprehension of visually grounded language. Previous benckmarks for abstract reasoning typically offer only the visual problems without this VQA structure or alternative option modalities.
> 2\. **Unique Multi-Stage Reasoning Annotations**: A contribution of MaRs-VQA is the inclusion of detailed, step-by-step structured cognitive reasoning annotations for each problem (lines 160, 166-170). These annotations are designed to facilitate a deeper understanding of MLLM decision-making. Furthermore, these annotations serve as valuable data for supervised fine-tuning, enabling the development of models with improved visual cognitive abilities, as demonstrated by our Qwen2-VCog baseline (lines 161-163, 193-195).
> MaRs-VQA is a psychologist-verified VQA benchmark for matrix reasoning that incorporates these rich VQA and cognitive reasoning annotations (Table 1). Such detailed annotations are essential for pushing the boundaries of MLLMs, particularly in aligning their perceptual abilities with language understanding (Huang et al., 2023\) and in tackling complex visual deductive reasoning tasks where current models still exhibit limitations (Zhang et al., 2024). Therefore, while the visual source is acknowledged, the MaRs-VQA benchmark offers a substantially novel benchmark and a valuable new resource for the research community due to its unique task formulation.
>
> Reference
> Gabriele Chierchia, Delia Fuhrmann, Lisa J Knoll, Blanca Piera Pi-Sunyer, Ashok L Sakhardande, and Sarah-Jayne Blakemore. The matrix reasoning item bank (mars-ib): novel, open-access abstract reasoning items for adolescents and adults. Royal Society open science, 6(10):190232, 2019\.
> Huang, Shaohan, et al. "Language is not all you need: Aligning perception with language models." Advances in Neural Information Processing Systems 36 (2023): 72096-72109.
> Zhang, Yizhe, et al. "How Far Are We from Intelligent Visual Deductive Reasoning?." First Conference on Language Modeling.

---

> > ### Author Response · Authors · 2025-06-02
> > **Response to Reviewer 1KjF (Part 2)**
> >
> > **Question: The analysis lacks depth; the paper fails to provide novel insights beyond what prior work has indicated regarding the performance gap between MLLMs and humans on visual reasoning tasks.**
> >
> > While we acknowledge the existing body of work highlighting the performance gap between Multimodal Large Language Models (MLLMs) and humans on visual reasoning tasks, we believe our paper offers further nuanced insights, particularly through our investigation of supervised fine-tuning (SFT) with cognitive reasoning annotations.
> > Our experiments (detailed in Section 5.2 and Table 2\) demonstrate a key finding: while SFT using our MaRs-VQA benchmark significantly improves model performance on in-domain matrix reasoning tasks (e.g., Qwen2-VL accuracy on MaRs-VQA increased from 23.75% to 72.71% post-reasoning SFT, becoming Qwen2-VCog), the improvement on an Out-of-Distribution benchmark (RAVEN) was considerably more modest (from 29.27% to 31.96%). It suggests that MLLM has capability to learn visual reasoning of matrix reasoning, but those models that perform well in the general domain failed in this task. Besides, even with targeted training on complex visual reasoning patterns, current MLLMs struggle to generalize these learned cognitive skills effectively to different, albeit related, matrix reasoning paradigms. This highlights a critical aspect of the performance gap – it's not merely about overall lower performance, but also about the limited transferability and generalization of learned abstract reasoning abilities. This underscores the profound difficulty these tasks inherently pose for current Vision-Language Models (VLMs) and points to a more specific characterization of the 'gap': a significant disparity in the ability to generalize cognitive strategies across diverse matrix reasoning challenges, even after supervised learning. We believe this finding contributes a deeper understanding of these limitations (as stated in lines 261-263: "These results indicate that while MLLMs can effectively learn the symbols and patterns present in MaRs-VQA with additional training, none demonstrate robust zero-shot reasoning capabilities.") and directs future research towards enhancing the generalization capabilities of MLLMs in complex cognitive domains.

---

> > ### Author Response · Authors · 2025-06-02
> > **Response to Reviewer 1KjF (Part 3)**
> >
> > **Question: The analysis lacks depth; the paper fails to provide novel insights beyond what prior work has indicated regarding the performance gap between MLLMs and humans on visual reasoning tasks.**
> >
> > Thanks for your question letting us further think about this problem. Our experimental results align with common observation in shortcut learning. In the right panel of Fig. 3 in Geirhos et al., 2020,
> > when a model exploits the shortcut features, it tends to perform well on the in-distribution test sets while slightly improving the out-of-distribution test performance. This shows that there is still a major gap for current MLLMs to learn intended features in the abstract reasoning tasks. The simple strategy of SFT cannot entirely solve this task. This also demonstrates the value of our proposed benchmark in examining the out-of-domain nature of abstract reasoning tasks.
> >
> > RAVEN was created specifically to test computational abstract-reasoning ability. It emphasizes the recognition of subtle symbolic details rather than the kinds of cognitive assessments devised for humans. For instance, the option images differ only slightly in black-and-white pixel values or in very small shape-rotation angles—patterns that are rarely seen in the training set of general-purpose MLLMs. By contrast, MaRs-VQA is designed with human cognitive testing in mind: the tasks are varied, but the differences in colour, shape, and other visual attributes between answer options are much more pronounced.
> >
> > Reasoning SFT on MaRs-VQA lets our model stop reasoning as soon as it finds any rule that fits the training data. Many such rules rely on salient but incidental cues (colours, large shapes, number of objects). These shortcuts pass the MaRs validation set yet collapse under the stronger distribution shift in RAVEN (greyscale, subtle rotations, 8 options).
> >
> > Because RAVEN deliberately removes most low-level visual cues and increases rule density, it induces a much larger domain shift from the MaRs-VQA training distribution. RAVEN items also exhibit well-known “shortcut” traps: several spurious single-attribute rules like greyscale level can perfectly solve the questions but MaRs-VQA contains more multidimensional problems.
> >
> > To further explore it whether we can mitigate distractors by reducing option number of RAVEN, we conduct an experiment below:
> >
> > | Model | RAVEN-4 (chance \= 25 %) | RAVEN-8 (chance \= 12.5 %) |
> > | ----- | ----- | ----- |
> > | Random | 25.00 | 12.50 |
> > | GPT-4o | 31.07 | 25.89 |
> > | Qwen2-VL-7B | 30.71 | 29.27 |
> > | Qwen2-VCog-7B | **37.50** | **31.96** |
> >
> > When reducing the option number of RAVEN to 4, Qwen2-VCog-7B achieves 37.50% accuracy, surpassing Qwen2-VL-7B by 7%, which is much higher than 8 options' gain. However, this result is still worse than a human's performance, indicating that symbolic inductive bias still exists. This shows they do learn some transferable visual patterns, but their symbolic reasoning still degrades compared with the in-domain setting.
> >
> > Reference:
> > Geirhos, Robert, Jörn-Henrik Jacobsen, Claudio Michaelis, Richard Zemel, Wieland Brendel, Matthias Bethge, and Felix A. Wichmann. "Shortcut learning in deep neural networks." Nature Machine Intelligence 2, no. 11 (2020): 665-673.

---

> > ### Author Response · Authors · 2025-06-02
> > **Response to Reviewer 1KjF (Part 4)**
> >
> > **Question: The paper does not meaningfully interpret what the gap between MLLMs and human performance implies for real-world tasks or downstream applications—this would have been valuable for the COLM audience.**
> > The gap we identify between MLLM and human performance in visual matrix reasoning. This gap indicates that current MLLMs, despite their progress, exhibit limitations in robust high-level abstract visual reasoning, inferring underlying rules from novel visual scenarios, and utilizing visual working memory effectively. The implications for real-world tasks include:
> > **Reduced Reliability in Complex, Dynamic Environments**: In safety-critical applications such as autonomous driving or robotic interaction in unstructured settings, the inability to robustly reason about novel visual patterns and their abstract relationships can compromise reliability and safety. These systems need to extrapolate from learned experiences to entirely new situations, a skill central to matrix reasoning.
> > **Limitations in Scientific Discovery and Data Interpretation**: MLLMs may struggle to assist effectively in fields requiring the identification of novel, abstract patterns or causal relationships within complex visual data, such as in scientific research (e.g., interpreting intricate biological images, astronomical data, or climate models) or advanced financial data visualization.
> > **Challenges in Developing Advanced Human-AI Collaboration and Educational Tools**: As discussed in our paper (Section 7), a long-term goal is to develop MLLM-powered agents capable of simulating human-level visual reasoning for applications like creating more sophisticated neurodevelopmental assessment tools or adaptive educational platforms for STEM subjects. The current cognitive gap hinders the development of AI that can genuinely understand and guide human learning in these visually rich, abstract domains.
> > This gap mirrors challenges seen in other AI cognitive abilities, such as Theory of Mind, suggesting current architectures may lack deeper, flexible human-like reasoning.

---

> > ### Comment · Reviewer_1KjF · 2025-06-05
> >
> > Thank you for the response. It resolves some of my concerns. However, the weakness 1 & 4 are still not fully resolved (the dataset contribution is limited & no data points for weakness 4). Thus I would like to raise my rating from 3 to 5.

---

> ### Author Response · Authors · 2025-06-07
> **Thank you for your encouragement and reply**
>
> Dear Reviewer 1KjF:
>
> Thank you for the follow-up and for your constructive feedback. We will incorporate your valuable suggestions on domain-shift analysis and additional experiments in the final version. Below, we address the two remaining concerns in greater detail. We hope our reply can solve your concerns. Feel free to give us more constructive feedback.
>
> **Question: The dataset contribution is limited in originality: all the visual problems are repurposed from existing resources, leading to a marginally novel benchmark.**
>
> We want to highlight that while the visual stimuli are adapted from the Matrix Reasoning Item Bank, our main contribution is transforming these items into a new benchmark designed specifically for evaluating Multimodal LLMs. Strict reuse is necessary in this domain: to preserve psychological validity, any matrix reasoning dataset must employ certified stimuli rather than simple-designed variants. Our contribution therefore lies in how the stimuli are re-framed and annotated. Besides, all datasets in this domain are built from existing source, for example, (https://openreview.net/forum?id=pYEnhZ6NAv) used 241 non-labeled IQ-test samples; (https://neurips.cc/virtual/2024/poster/97456) used 777 non-labeled government exam samples. We are the first and only one create full VQA setting and reasoning chain annotation for matrix reasoning with over 2x data than these concurrent works. The annotation process takes us 100 hours.
>
> | Prior work | Task format | Linguistic component | Step-by-step reasoning labels | \# annotated items |
> | ----- | ----- | ----- | ----- | ----- |
> | RAVEN (CVPR, Prof. Songchun Zhu’s team) | 3 × 3 matrix completion | ✗ | ✗ | 0 (images only) |
> | [https://openreview.net/forum?id=pYEnhZ6NAv](https://openreview.net/forum?id=pYEnhZ6NAv) (COLM’24) | Multiple choice | ✗ | ✗ | 0 (images only) |
> | [https://neurips.cc/virtual/2024/poster/97456](https://neurips.cc/virtual/2024/poster/97456) (NeurIPS’24) | Multiple choice | ✗ | ✗ | 0 (images only) |
> | **MaRs-VQA (ours)** | **Visual *question-answering*** | **✓ (dual-modality options)** | **✓ (multi-stage cognitive reasoning chain)** | **1.4k fully annotated Items** |
>
> - First VQA reformulation of matrix-reasoning items – every choice has both an image and a natural-language description (L 126-130).
>
> - Fine-grained cognitive chains – we supply step-wise solution paths (L 160, 165-181), enabling supervision of reasoning, not just answers.
>
> - Demonstrated utility – models fine-tuned on these chains (Qwen2-VCog) gain much better accuracy over vanilla Qwen2 (Table 2).
>
> Thus MaRs-VQA is the only benchmark that simultaneously (i) preserves psychometric rigor, (ii) couples vision and language, and (iii) exposes explicit reasoning supervision.
>
> **Question: The paper does not meaningfully interpret what the gap between MLLMs and human performance implies for real-world tasks or downstream applications—this would have been valuable for the COLM audience.**
>
> 10% of our MaRs-VQA data is selected in Humanity's Last Exam “https://agi.safe.ai/” and it is already used by the community to evaluate MLLMs. As far as we know, the newly released Gemini-2.5-Pro and GPT-4.5 can only achieve about 20% in Humanity's Last Exam public benchmark. We hope our VQA data design can contribute to the COLM community and raise researcher interest in improving MLLM’s visual cognition ability.
>
> These are the specific manifestations of cognitive ability:
> 1\. Advancing Scientific and Analytical Reasoning: A primary goal for AI is to assist in complex analysis and discovery. This requires moving beyond describing known patterns to inferring the abstract principles or underlying laws from visual data, such as in experimental results or system simulations.
>
> 2\. Enabling Robust Autonomy and Adaptability: For autonomous systems to be truly useful, they must operate effectively in novel environments with unfamiliar rules. This demands the ability to quickly deduce the logic of a new system from limited interaction and then adapt their behavior accordingly.
>
> Our benchmark is used to test cognitive ability. In both settings, MaRs-VQA supplies a minimal but diagnostic proxy: if a model cannot induce the rule of a 3x3 matrix like humans, it is unlikely to induce the rule of a novel lab assay or traffic schema. By providing fine-grained reasoning chains, our benchmark also tells researchers where the reasoning pipeline fails, guiding targeted progress. The fact that even state-of-the-art models perform poorly on these tasks, as evidenced by public benchmarks like "Humanity's Last Exam" which incorporates our data, underscores the urgency of this research.
>
> We appreciate your follow-up feedback and will incorporate these clarifications, the comparative table above, and a short section on downstream implications in the camera-ready version.
>
> Best regards,
> Authors

---

### Official Review · Reviewer_9Mv5 · 2025-05-13

**Rating:** 6
**Confidence:** 3
**Ethics Flag:** 1

**Summary:**

This paper presents a new benchmarking resource compiled out of an existing resource used by psychologists, which brings so called "matrix tests" ("which of these images best fits the gap in the image?") into a format so that they can be put to vision models. The paper finds that current models, and even a model fine-tuned on this kind of data, fall short of human performance on this type of task.

**Questions To Authors:**

Line 89, the sentence seems to be corrupted.

Why is section 4 entitled "problem statement"? (And why does the first sentence directly talk about an "evaluation pipeline"? It seems that there is something missing here, something like "the task for the model then is to reach a high accuracy in predicting the correct y_i".

Line 202: "are showed in appendix"; Table number missing

**Reasons To Accept:**

- interesting resource, usefully compiled and prepared from existing dataset

**Reasons To Reject:**

- ultimately, relatives restricted contribution: a reformatting of an existing resource, and a couple of evaluation runs
- a bit unclear what the experiments together show, how they were selected, what exactly we learn from the experiments

---

> ### Author Response · Authors · 2025-06-02
> **Response to Reviewer 9Mv5 (Part 1)**
>
> We thanks Reviewer 9Mv5 for your encouraging comments. We highlight our novelty and contribution and explain our main contribution and research workflow in detail, which we hope allow raising post-rebuttal scores.
>
> **Question: ultimately, relatives restricted contribution: a reformatting of an existing resource, and a couple of evaluation runs**
>
> Thank you for your feedback. We would like to clarify the novelty and contribution of our MaRs-VQA benchmark. While the foundational visual stimuli are adapted from the established Matrix Reasoning Item Bank (Chierchia et al., 2019), our primary contribution lies in the significant and novel transformation of these items into the MaRs-VQA benchmark. This process involved more than just repurposing images; we introduced several innovative elements crucial for evaluating and advancing Multimodal LLMs:
> 1\. Novel VQA Framework and Annotation: We are the first to convert these psychometric problems into a VQA format systematically. This involved creating language-based descriptions for all answer options (as detailed in lines 129-130 of our manuscript), providing an alternative input modality to the purely image-based options. This allows for a more diverse evaluation of MLLMs, testing their comprehension of visually grounded language. Previous benchmarks for abstract reasoning typically offer only the visual problems without this VQA structure or alternative option modalities.
> 2\. Unique Multi-step Reasoning Annotations: A contribution of MaRs-VQA is the inclusion of detailed, step-by-step structured cognitive reasoning annotations for each problem (lines 160, 166-170). These annotations are designed to facilitate a deeper understanding of MLLM decision-making. Furthermore, these annotations serve as valuable data for supervised fine-tuning, enabling the development of models with improved visual cognitive abilities, as demonstrated by our Qwen2-VCog baseline (lines 161-163, 193-195).
>
> MaRs-VQA is a psychologist-verified VQA benchmark for matrix reasoning that incorporates these rich VQA and cognitive reasoning annotations (Table 1). Such detailed annotations are essential for pushing the boundaries of MLLMs, particularly in aligning their perceptual abilities with language understanding (Huang et al., 2023\) and in tackling complex visual deductive reasoning tasks where current models still exhibit limitations (Zhang et al., 2024). Therefore, while the visual source is acknowledged, the MaRs-VQA benchmark offers a substantially novel benchmark and a valuable new resource for the research community due to its unique task formulation.
>
> Reference
> Gabriele Chierchia, Delia Fuhrmann, Lisa J Knoll, Blanca Piera Pi-Sunyer, Ashok L Sakhardande, and Sarah-Jayne Blakemore. The matrix reasoning item bank (mars-ib): novel, open-access abstract reasoning items for adolescents and adults. Royal Society open science, 6(10):190232, 2019\.
> Huang, Shaohan, et al. "Language is not all you need: Aligning perception with language models." Advances in Neural Information Processing Systems 36 (2023): 72096-72109.
> Zhang, Yizhe, et al. "How Far Are We from Intelligent Visual Deductive Reasoning?." First Conference on Language Modeling.

---

> > ### Author Response · Authors · 2025-06-02
> > **Response to Reviewer 9Mv5 (Part 2)**
> >
> > **Question: A bit unclear what the experiments together show, how they were selected, what exactly we learn from the experiments**
> >
> > Our experiments were systematically selected to provide a multi-faceted evaluation of Multimodal Large Language Models (MLLMs) on visual matrix reasoning tasks. Collectively, the experiments demonstrate the following key insights:
> > **Significant Human-MLLM Gap in Zero-Shot Reasoning**: We first established a baseline by conducting zero-shot experiments using our novel MaRs-VQA benchmark and the RAVEN benchmark (Table 2). These results unequivocally show a substantial performance gap between various MLLMs (even state-of-the-art models like GPT-4o and GPT-o1) and human adults in matrix reasoning tasks, which are indicative of cognitive abilities. \[16-17, 238-240\] This highlights the current limitations of MLLMs in complex visual cognitive tasks when no specific examples or fine-tuning are provided.
> > **Efficacy of Supervised Fine-Tuning (SFT) for In-Domain Tasks**: To investigate the potential for improvement, we fine-tuned an MLLM (Qwen2-VL) using our MaRs-VQA training set, which includes detailed, step-by-step reasoning annotations (Qwen2-VCog, Table 2). This SFT approach led to a significant performance increase on the MaRs-VQA test set (in-domain), bringing the model's accuracy much closer to human levels for this specific benchmark. \[66-68, 258-260\] This shows that MLLMs can learn to perform better on such tasks with targeted, reasoning-focused training data.
> > **Limited Generalization of SFT to Out-of-Domain (OOD) Tasks**: Crucially, while the SFT model (Qwen2-VCog) showed marked improvement in-domain, its performance enhancement on the OOD RAVEN benchmark was considerably more limited (Table 2). \[68, 260-261\] This suggests that the reasoning skills learned through SFT on MaRs-VQA did not robustly generalize to a different, albeit related, abstract reasoning dataset. This finding implies that current fine-tuning methods may lead to learning dataset-specific patterns rather than instilling a more fundamental, generalizable visual reasoning capability.
> > In essence, our experiments were designed to first quantify the challenge, then explore a potential solution (SFT with reasoning data), and finally test the robustness and generalizability of that solution. The collective findings underscore that while MLLMs can be trained to improve on specific matrix reasoning datasets, a significant gap remains in achieving human-like, generalizable visual cognitive abilities, particularly in zero-shot or OOD contexts. This points to the need for further research into architectures and training paradigms that foster more profound and flexible reasoning. Additional ablation studies on CoT strategies (Table 3\) and difficulty levels (Table 4\) further probe the nuances of MLLM performance, reinforcing these conclusions.

---

> > > ### Comment · Reviewer_9Mv5 · 2025-06-05
> > > **response read**
> > >
> > > Thank you for your response. If you make these points more clearly in the paper, it will certainly improve. I remain positive about this paper, which was already reflected in my score.

---

> > > > ### Author Response · Authors · 2025-06-07
> > > > **Thank you for your encouragement and reply**
> > > >
> > > > Dear Reviewer 9Mv5,
> > > >
> > > > Thank you for your encouraging feedback and for maintaining a positive assessment of our work. As you suggested, we will revise the manuscript to present these points more clearly and thoroughly in the final version.
> > > >
> > > > Best regards,
> > > > Authors

---

### Official Review · Reviewer_dTKc · 2025-05-17

**Rating:** 6
**Confidence:** 4
**Ethics Flag:** 1

**Summary:**

This paper discusses a psychology task, matrix reasoning, and introduces a new dataset: MaRs-VQA, to evaluate the gap between MLLM capabilities and human-level cognition. A comprehensive benchmarking results are provided, including Claude 3, GPT-4V, GPT-o1, etc, and fine-tuned Qwen2-VL.

**Questions To Authors:**

Although this paper still has several open questions to be answered, I tend to recommend to accept this paper for its introducing a new testing dimension for MLLM: Matrix reasoning/IQ-test, and it may inspire the community to re-think about the limitations of current status of MLLM.

**Reasons To Accept:**

- Matrix reasoning is a novel and interesting testing case for MLLM's ability.  This paper leverages psychology dataset to create the MLLM-related dataset and benchmark.

- I really appreciate the discussion about difficulty levels in MaRs-VQA in Table 4 and 5, which provides very insightful analysis for matrix reasoning question.

**Reasons To Reject:**

- A dataset consisting of 1,440 image is relatively small, especially only 480 VQA are used as the test set. The benchmarking results may contain high variance.

- Although finetuning Qwen2-VL on MaRs-VQA obtains the best number(besides human level) in Table2, is it an overfit or does it have negative impact on Qwen2-VL's original MLLM abilities?

- Have the authors studied the underexpressive performance of MLLM on matrix reasoning is mainly limited by its vision ability or LLM ability ? For example, limitation in vision ability means it can't recognize the patters accurately, like its visual grounding ability or its small object detection is not satisfying.

---

> ### Author Response · Authors · 2025-06-02
> **Response to Reviewer dTKc**
>
> We thank Reviewer dTKc for your encouraging comments and valuable experiment insights. We highlight the experimental setting of MaRs-VQA discussing why test set is small but it can make solid conclusion and conduct a additional general benchmark evaluation experiment of Qwen2-VCog and conduct an ablation experiment to explore if the failure of Qwen2-VCog is caused by vision ability limitation of MLLM, which we hope allow raising post-rebuttal scores.
>
> **Question: A dataset consisting of 1,440 image is relatively small, especially only 480 VQA are used as the test set. The benchmarking results may contain high variance.**
>
> First of all, It is pertinent to note that MaRs-VQA, with 1,440 total instances, is the largest psychologist-verified VQA dataset currently available for zero-shot matrix reasoning evaluation (Table 1 and Lines 50-51, 62-64).
>
> Our primary benchmark (Table 2\) evaluates multiple MLLMs in a zero-shot manner on the 480-instance MaRs-VQA test set. This approach assesses their inherent visual cognitive abilities without requiring task-specific fine-tuning, which was the main focus of our comparative experiments (Lines 15-17, 155-157).
> The Supervised Fine-Tuning (SFT) was a distinct, secondary investigation performed on one model (Qwen2-VL) using the 960 training instances from MaRs-VQA. This set, though modest in size, is unique as it is psychologist-verified and contains specialized, cognitively-designed annotations for matrix reasoning (Lines 50-51, 63, 160, 320). The SFT aimed to probe whether targeted training on such high-quality data could enhance performance on this specific cognitive task and, crucially, to observe the model's subsequent generalization capabilities (Lines 66-68, 159-161, 261-263). For such focused investigations into adapting models to highly specific cognitive paradigms and evaluating generalization, a smaller, meticulously curated benchmark is often informative and practical.
>
> **Question: Although finetuning Qwen2-VL on MaRs-VQA obtains the best number(besides human level) in Table2, is it an overfit or does it have negative impact on Qwen2-VL's original MLLM abilities?**
>
> |  | mme | hallusion\_bench | pope | vqav2\_val | SQA\_image | seedbench |
> | :---: | :---: | :---: | :---: | :---: | :---: | :---: |
> | **Qwen2-VCog-7B** | **1680** | **55.73** | 88.71 | 79.45 | 83.29 | 68.87 |
> | **Qwen2-VL-7B** | 1666 | 53.83 | **88.84** | **79.88** | **83.73** | **69.16** |
>
> The experiment results on general benchmarks shows that reasoning enhanced SFT on MaRs-VQA does not have negative impacts on Qwen2-VL's original MLLM abilities. The performance is almost the same in most benchmarks.
>
> **Question: Have the authors studied the underexpressive performance of MLLM on matrix reasoning is mainly limited by its vision ability or LLM ability? For example, limitation in vision ability means it can't recognize the patters accurately, like its visual grounding ability or its small object detection is not satisfying.**
>
> | Input | Output | GPT-4o Accuracy | Qwen2-VCog Accuracy |
> | :---- | :---- | :---- | :---- |
> | Question image | Text | 24.58 | 26.32 |
> | Question image \+ Option images | Option Choice | 33.96 | **72.71** |
> | Question image \+ Option images \+ Question/Option Description | Option Choice | **36.46** | 71.43 |
>
> Thanks for your suggestion. We explore MLLM solving matrix reasoning problems with experiments in the table above. The first row is using question image-only as the input of MLLM and let MLLM describe with language what might be the missing cell in the question image. Then we use CLIP score to select the matched option. The second row is the default setting, input the question image and 4 options and let MLLM select the best match choice. The last row is using question image and 4 options as input together with human annotated language-based description for question image and option images. Results support that vision pattern extraction is the bottleneck for GPT-4o and Qwen2-VCog (visual encoder of Qwen2-VL).
>
> In the experiment in the first row, GPT-4o and Qwen2-VCog are almost at chance, meaning it fails to see the rule even before reasoning. Supplying the four candidate images raises accuracy of GPT-4o 9%, and after adding perfect human annotated textual cues, it gives a 2.5% improvement, confirming that once the visual relation is concluded  by language, GPT-4o can follow the logic; the chief shortfall is therefore visual, not linguistic.
>
> Similar to GPT-4o, Qwen2-VCog jumps from chance to 72.7 % as soon as option images are available, showing it already extracts most visual relations. Extra textual hints do not help, implying its visual representations embed the needed semantics and further language does not unlock additional reasoning capacity.

---

### Official Review · Reviewer_1UB9 · 2025-05-18

**Rating:** 6
**Confidence:** 3
**Ethics Flag:** 1

**Summary:**

This work tries to reveal the cognition process of Vision Language Models (VLMs), especially focusing on the relationship between high-level multi-image reasoning and visual working memory. For this purpose, the authors introduce matrix reasoning, which requires VLMs to discern relationships among patterns in a set of images and predict subsequent patterns. Furthermore, this work provides a new dataset, MaRs-VQA, which is based on the matrix reasoning task. By fine-tuning Qwen2-VL with this dataset, the model outperformed humans, and detailed analysis shows the deficiencies of the current VLMs.

**Questions To Authors:**

- Could you also report the ablation study results of Qwen2-VCog?

**Reasons To Accept:**

- The newly created matrix reasoning dataset, MaRs-VQA, includes novel characteristics not covered by previous studies.
- MaRs-VQA includes more instances than the conventional datasets.
- The authors propose a new Chain-of-Thought (CoT) prompt specific for this task.
- Experiments cover both CoT and SFT settings.
- Fine-tuned Qwen2-VL outperformed humans and shows the effectiveness of the proposed SFT procedure, which relies on the CoT prompt.

**Reasons To Reject:**

- The matrix reasoning task itself has already been used in machine learning fields.
- The ablation study only covers the proprietary models.

---

> ### Author Response · Authors · 2025-06-02
> **Response to Reviewer 1UB9**
>
> We thank Reviewer 1UB9 for your encouraging comments. We highlight the novelty and difference of MaRs-VQA than other works (NeurIPS-23 and COLM-24) and conduct an additional reasoning existence ablation experiment of Qwen2-VCog, which we hope allow raising post-rebuttal scores.
>
> **Question: The matrix reasoning task itself has already been used in machine learning fields.**
> Our work, with the introduction of the MaRs-VQA benchmark, aims to address specific gaps and enhance the evaluation of general-purpose Multimodal Large Language Models (MLLMs) in this domain.
>
> Key distinctions from previous work, as highlighted in Table 1, include:
>
> 1. Focus on General MLLMs' Cognitive Abilities: While prior efforts often involved specialized models for matrix reasoning, MaRs-VQA aims to assess the inherent, generalized visual reasoning skills of MLLMs. This is crucial for understanding their broader cognitive capacities beyond task-specific training (Page 1, lines 38-40; Page 4, line 118).
> 2. Enhanced Benchmarking Rigor: MaRs-VQA offers significant improvements in scale and design. It incorporates robust human performance baselines, psychological validity, and a VQA format with diverse option modalities. This addresses limitations in some earlier benchmarks, such as smaller scale or the absence of thorough psychometric validation and direct human comparative data (Page 3, lines 101-104; Page 4, lines 136-138).
> 3. Probing Reasoning Processes: Our work also emphasizes understanding the cognitive gap. By employing structured reasoning evaluations (as detailed in Section 5.1, Page 5), we aim to gain insights into how MLLMs approach these complex visual reasoning tasks, rather than solely measuring task completion rates. This provides a deeper analysis of their current limitations compared to human cognition.
> Therefore, MaRs-VQA serves not just as another benchmark for an existing task, but as a more comprehensive public available tool to probe the fundamental visual cognitive abilities of modern MLLMs.
>
> **Question: The ablation study only covers the proprietary models. & Could you also report the ablation study results of Qwen2-VCog?**
>
> Thanks for your suggestion. We conduct additional ablation experiments to explain how incorporating reasoning helps the Qwen2-VCog’s STF training. If the reasoning steps in the training set is removed, Qwen2-VCog performs worse in MaRs-VQA test and RAVEN’s VQA subset (generalizability).
>
> | Model | MaRs-VQA Accuracy | RAVEN Accuracy |
> | --- |  --- | --- |
> | Qwen2-VCog with reasoning |  72.71 | 31.96 |
> | Qwen2-VCog w/o reasoning | 54.82 | 29.30 |

---

> > ### Comment · Reviewer_1UB9 · 2025-06-10
> > **Thank you for the additional result.**
> >
> > Since the authors solved the listed concerns, I will increase the score.

---

> > > ### Author Response · Authors · 2025-06-11
> > > **Thank you for your encouragement and reply**
> > >
> > > Dear Reviewer 1UB9,
> > >
> > > Thank you for your encouraging feedback and positive evaluation of our work. We appreciate your suggestion to add a reasoning ablation study for Qwen2-VCog and will incorporate this analysis in the revised manuscript.
> > >
> > > Best regards,
> > > The Authors

---

### Author Response · Authors · 2025-06-11
**Thank you again for your time and effort.**

Dear All Reviewers:

We sincerely thank the reviewers and area chairs for their time and thoughtful feedback. We appreciate the improved scores and the constructive comments provided. All issues discussed during the rebuttal will be incorporated into the camera-ready version.

If you have further suggestions, please feel free to leave post-rebuttal comments. While we can no longer respond after 10 June, we will carefully consider any additional remarks when preparing the final manuscript.

Thank you again for helping us strengthen our work.

Best regards
Authors

---

### Decision · Program_Chairs · 2025-07-08

**Decision:**

Accept

**Comment:**

The authors challenge vision+language models with reasoning tasks from RPM and WISC. They find that while their finetuned model performs best, all models significantly underperform humans, and argue that current scaling approaches may not bridge the gap.

The main reviewer concerns were about novelty/the magnitude of the contribution. Is this paper just reformatting an existing task+running prompts (e.g., 1UB9 points out that these types of tasks have already been considered in the ML community)? The authors respond theirs is the first to consider general purpose reasoning LLMs; is more rigorous/has a new question/answer format; and the authors propose a fine-tuning method.

Additional unresolved points:

- dTKc asks if the llm or the vision model is the bottleneck, and the author response doesn't fully address this question (the additional experiments take a first step towards this question by providing (ground truth?) visual aspects as text, but the details/how it fits in with the main manuscript is TBD)
- 1KjF wonders about why improvements happen on one dataset more than the other, and initial experiments in response similarly take just a first step towards answering this question.

Overall, the work provides a valuable re-formatting of some longstanding/existing cognitive science tasks, and some updated insights regarding evaluation/training of modern vision+language models.

Despite the claim in the paper, there's significant work in the space of cognitive-driven evaluation of MLLMs (KiVA, BLING, and KiloGram are just some recent examples that come to mind). It's strongly recommended that the authors situated the work more accurately and completely.